# Research on Aftertreatment Inlet_Outlet Insulation for A Nonroad Middle Range Diesel Engine

**Lu Xie [1], Guozhang Jiang [1] and Feng Qian [2],***

[1] College of Machinery and Automation, Wuhan University of Science and Technology, No. 947 Heping Road, Wuhan 430081, Hubei, China; xiemenglu@163.com (L.X.); whjgz@wust.edu.cn (G.J.)

[2] College of Automotive and Transportation Engineering, Wuhan University of Science and Technology, No. 947 Heping Road, Wuhan 430081, Hubei, China

* Correspondence: feng.qian@wust.edu.cn

**Abstract:** Diesel exhaust aftertreatment systems are required for meeting China StageIV emission regulations. This paper addresses an aftertreatment system designed to meet the China StageIV emission standards for nonroad vehicle markets. It presents a comprehensive experimental research work on aftertreatment skin temperature and the radiated impact on its neighboring parts in a nonroad vehicle powered by a middle range diesel engine under aftertreatment inlet/outlet with insulation and without insulation with multiple experimental conditions, as well as validating the emission results with these two different aftertreatment configurations. According to the experimental results, it can be observed that the aftertreatment inlet/outlet with insulation and without insulation using a Diesel Oxidant Catalyst (DOC) + Diesel Particle Filter (DPF) + Selective Catalytic Reduction (SCR) scheme could both meet China StageIV emission regulations and the whole vehicle arrangement. The connection pipe is generally short between the aftertreatment and the engine turbo charger on nonroad application vehicles, which results in the exhaust gas temperature of the internal aftertreatment at each point being similar, with variation within ±2% for the aftertreatment inlet/outlet with insulation compared to the aftertreatment inlet/outlet without insulation. The aftertreatment skin temperature differences under these two configurations occur on the inlet module and outlet module, and the skin temperatures of other aftertreatment modules are little impacted. These experimental results also validate the radiation model. All aftertreatment skin temperatures are measured with different experimental conditions. In future, if considering integrating other parts like sensors on the surface of the aftertreatment, the configuration with insulation is recommended. As per the experimental results, the maximum inlet skin temperature can lower nearly 50% with insulation and the maximum outlet temperature could lower about 28% compared to the configuration without inlet/outlet insulation. If taking cost into consideration, the configuration without insulation is suggested. This research also introduces alternative solutions for different concerns for real applications. The methodology provides effective guidance and reference for future aftertreatment insulation considerations for inlet modules and outlet modules on real applications.

**Keywords:** aftertreatment; insulation; skin temperature; experimental

## 1. Introduction

China's recently proposed StageIV emission regulations preliminarily stated that the system NOx emission limit for a middle range diesel engine(power within 130 kW–560 kW) should be 2.0 g/kWh, the $NH_3$ slip limit 25 ppm, and the Particle Matter(PM) limit 0.025 g/kWh, while a Particle Number(PN) requirement is also added with a limitation of $5 \times 10^{12}$ [1]. Fulfilling strict

emission regulations on nonroad diesel engine requires a complex aftertreatment with DOC + DPF + SCR. There are a lot of studies on the exhaust gas temperature impact on the aftertreatment performance of catalysts. Evan et al. studied that exhaust gas temperature would affect catalyst conversion efficiency [2]. Johnson and Kim et al. showed that the aftertreatment catalyst light-off would be delayed by a considerable decrease in exhaust gas temperatures. Thus, it is very important to manage the gas temperature during aftertreatment to make sure the catalysts perform well under appropriate temperatures [3,4]. Ramesh et al. demonstrated the thermal management benefit of complex aftertreatment temperature control in a certain system temperature range, which could meet stringent emission regulations. Aftertreatment temperature is a significant consideration to achieve the best conversion efficiency [5]. There are also many researchers who have made efforts to study and optimize the exhaust gas temperature on aftertreatment. Joshi and Serrano et al. used engine design optimizations to ensure a particular aftertreatment component temperature [6,7], and Hamedi indicated that it is possible to implement a thermal energy storage system to ensure the exhaust gas temperature for catalyst conversion efficiency is compliant with critical emission levels [8]. Meng and Bai et al. emphasized that the exhaust temperature is critical for DPF regeneration, and they applied experimental methods to optimize engine thermal management control strategies to achieve a specific exhaust gas temperature for DPF on PN control to meet the strict PN emission levels [9,10]. Luján and Holmer et al. applied engine calibration to control the aftertreatment inlet exhaust gas temperature and used modeling and analytical methods on the literature regarding what is critical for aftertreatment in order to reach the specific exhaust gas temperatures for the best conversion efficiency [11,12] Aftertreatment packaging design is another key area of research in existing studies. Liu et al. emphasized the importance of the diesel exhaust aftertreatment system design, especially the SCR system related packaging design, which affects SCR emission performance [13]. Computational fluid dynamics (CFD) and experimental methods are the main methodologies used to conduct the related studies. Hamedi et al. set up CFD models to simulate DOC emission conversion with different aftertreatment insulation strategies. The results showed that different insulation strategies will result in different DOC emission performances [14]. Kandylas et al. applied CFD simulation and dyno testing methods to provide optimization of the exhaust system design parameters. Different types of pipe insulation were the important factor when considering heat transfer in the exhaust pipe, which impacts the exhaust gas temperature performance to emissions results. It was demonstrated that insulation has a strong relationship with heat transfer and affects the related temperature [15]. Philip et al. used the simulation method for aftertreatment packaging design [16]. Wurzenberger and Johann et al. established models for simulating a comprehensive framework on different wall layers to analyze exhaust gas temperature under different operating conditions with the heat transport models [17,18].

These researches have demonstrated that exhaust gas temperature affects the performance of aftertreatment catalysts (DOC and SCR) while investigating the influence of different parameters on exhaust gas temperature. It is suggested add insulation on the surface of a DOC + DPF + SCR scheme when designing aftertreatment packaging, to try to lower the heat loss. Nonroad vehicles provide space for aftertreatment installation which is near to the engine using a shorter connection pipe between the engine turbo charger and aftertreatment inlet, which means that the exhaust gas temperature on aftertreatment inlet consists of the turbo charger out temperature with an insulated connection pipe between the turbo and aftertreatment inlet under certain duty cycles with robust electronic control module (ECM) calibration. In order to introduce aftertreatment with technical robustness as well as economic considerations, this research studies the overall temperatures at the surface of every aftertreatment component using two different aftertreatment inlet/outlet insulation strategies under multiple experimental conditions on a nonroad vehicle and analyzes the radiated heat impact to the neighbor parts on the vehicle. On the other hand, this research also introduces alternative solutions for different concerns for real applications, as well as validating the methodology, which provides effective guidance and reference for the consideration of future aftertreatment insulation for inlet and outlet modules on real applications. This paper aims to research aftertreatment inlet insulation and outlet

insulation on nonroad vehicles with diesel engines focusing on aftertreatment skin thermal mapping and radiated heat influence on the vehicle using experiments with real vehicles, while using a DOC + DPF + SCR aftertreatment scheme. In this study, two different packaging designs for aftertreatment inlet and outlet insulation strategies are provided by analyzing aftertreatment skin temperature and radiated heat impact to neighboring parts in a middle range diesel powered engine for nonroad vehicle marketing.

## 2. Results and Discussion

### 2.1. Experimental—Emission Result

Two different packaging designs for aftertreatment inlet and outlet insulation strategies are provided in this paper for aftertreatment skin temperature and radiated heat impact study, with the precondition that each strategy has met emission requirements. The research is conducted using aftertreatment architecture —DOC + DPF + SCR, and the two inlet and outlet insulation strategies are

(1) Strategy 1: Inlet with insulation and outlet with insulation;

(2) Strategy 2: Inlet without insulation and outlet without insulation.

The aim of this experimental part is to compare the exhaust gas temperature and nonroad transient cycle (NRTC) emission results with the two different packaging designs.

#### 2.1.1. Exhaust Gas Temperature Variation

Exhaust gas temperature is a very important parameter when evaluating the performance of the whole aftertreatment system. The average exhaust gas temperature at points on DOC_In, DPF_In, DPF_Out and SCR_Out are in the range of 250 °C to 265 °C with the two aftertreatment inlet/outlet insulation strategies under the same engine running conditions. Figure 1 indicates the average exhaust gas temperature variation between these two aftertreatment inlet/outlet insulation strategies under the same engine running conditions. In this figure, the delta temperature is calculated using the average temperature when the aftertreatment inlet/outlet has insulation minus the average temperature when the aftertreatment inlet/outlet does not have insulation. It shows the gas temperature variation is very small and in the range of ±5 °C at each considered aftertreatment system point, including DOC_In temperature, DPF_In temperature, DPF_Out temperature and SCR_Out temperature. The exhaust gas temperature variation ratio is within ±2%. It could be regarded that the exhaust gas temperature is little affected by the aftertreatment inlet/outlet insulation strategy with the nonroad application.

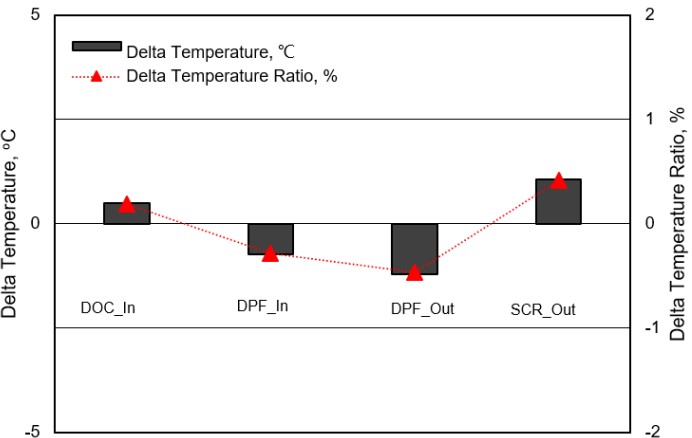

**Figure 1.** Exhaust gas temperature variation.

#### 2.1.2. Nonroad Transient Cycles (NRTC) Emission Results

It could be estimated that the emission performance would be similar for aftertreatment inlet/outlet with insulation and inlet/outlet without insulation as the exhaust gas temperature is very close at the

same point under the two aftertreatment configurations, if all other settings are consistent. Figure 2 shows aftertreatment system emission results under nonroad transient cycles and indicates that the whole system NOx emission is lower than 1.7 g/kWh, with the NOx conversion efficiency being about 84%, and the NH$_3$ slip is very low under both aftertreatment inlet/outlet insulation strategies with insulation and without insulation, which meets the nonroad China StageIV proposed regulation emission limit requirements, as does the PM emission and PN emission.

$$\eta = \frac{a_{in} - a_{out}}{a_{in}} \times 100\% \tag{1}$$

where $a_{in}$ is NOx emission from the engine out into aftertreatment, $a_{out}$ is NOx emission from aftertreatment out, and $\eta$ is NOx emission conversion efficiency.

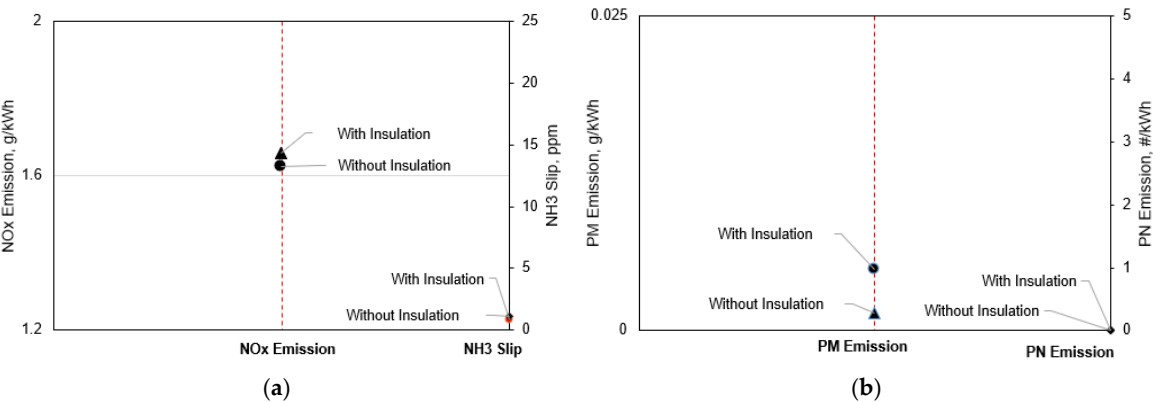

**Figure 2.** Nonroad transient cycles (NRTC) emission results. (**a**) NOx Emission and NH$_3$ Slip Results, (**b**) PM Emission and PN Emission Results.

The experimental results demonstrate that the aftertreatment emission could meet China StageIV regulation emission requirements with a margin, even using aged catalysts, whether with aftertreatment inlet/outlet insulation or not, which provides a reasonable precondition for aftertreatment skin temperature and radiated impact research.

### 2.2. Experimental—Aftertreatment Skin Temperature and Radiated Impact

#### 2.2.1. Theoretic Considerations

When introducing the aftertreatment into real applications, the aftertreatment skin temperature and radiated impact need to be taken into consideration. On the one hand, this helps to understand aftertreatment skin temperature mapping to provide guidance for safety, as well as taking into account the additional parts on the surface of aftertreatment with material having special temperature limit requirements. On the other hand, the radiated impact of aftertreatment on neighboring parts of the vehicle may need to be referenced when comprehensively designing the arrangement of vehicle parts. Although the aftertreatment inlet/outlet insulation strategies could both meet emission requirements, the aftertreatment skin temperature needs further research with and without insulation due to heat transfer.

NOx conversion and DPF regeneration are the significant processes when taking the two different aftertreatment inlet/outlet insulation strategies into considerations. These processes result in several conversions inside the aftertreatment by catalysts with heat influences.

Urea dosing into the mixer through a urea injector may be necessary according to a control strategy to achieve a specific NOx conversion efficiency in certain conditions to meet strict regulation requirements. Figure 3 [19] illustrates a urea pyrolysis mechanism and schematic for droplet evaporation, which explains the complex reactions during heat transfer stages within the aftertreatment. The urea

droplets begin to evaporate when the temperature rises due to being gradually heated by exhaust gas. The water starts to evaporate from the aqueous urea solution at the earliest point in time due to it having a lower boiling point. Urea can be directly pyrolyzed from the solid or liquid phase, so the second stage in Figure 3 is negligible.

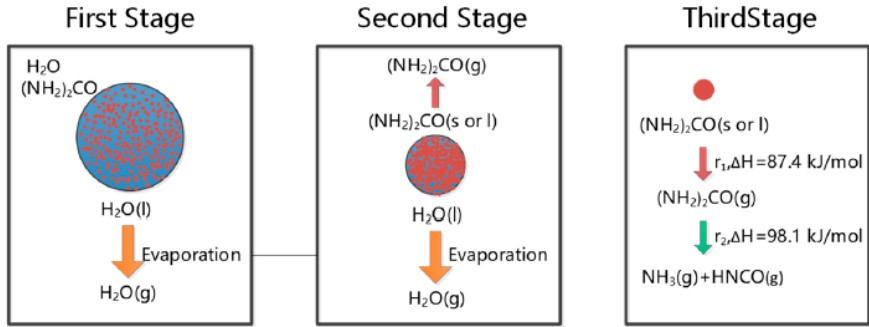

**Figure 3.** Urea pyrolysis mechanism [19]

In a word, the urea droplets are heated up and water evaporates first, followed by the thermolysis of urea into ammonia and isocyanic acid [20].

$$(NH_2)_2\,CO(aq)\ \rightarrow (NH_2)_2CO(1) + H_2O(g) \tag{2}$$

$$(NH_2)_2\,CO(1)\ \rightarrow NH_3(g) + NHCO(g) \tag{3}$$

The main reaction of NOx conversion is as below.

$$NOx(g) + NH_3(g)\ \rightarrow N_2(g) + H_2O(g) \tag{4}$$

DPF regeneration generally includes passive regeneration and active regeneration. Passive regeneration typically depends on the engine conditions and availability of NOx, which is the soot oxidization in the course of normal conditions as a result of the reaction of carbon with $NO_2$, which is from the NO oxidation in DOC.

$$2NO(g) + O_2(g)\ \rightarrow NO_2(g) \tag{5}$$

$$C(sol) + 2NO_2(g)\ \rightarrow CO_2(g) + 2NO(g) \tag{6}$$

Active regeneration actively increases the temperature to accelerate soot oxidization by $O_2$, achieved by engine fuel post injection to inject additional fuel and burning over DOC to achieve the regeneration temperature, which is higher than 500 °C. The thermal management in the whole system control strategy ensures this can be smartly realized.

$$HC(g) + O_2(g)\ \rightarrow H_2O(g) + CO_2(g) \tag{7}$$

$$C(sol) + O_2(g)\ \rightarrow CO_2(g) \tag{8}$$

The aftertreatment inlet is designed before the DOC module, while the outlet is after SCR. Considering multiple complex reactions occur in catalysts satisfying requirements under different conditions, the heat transfer onto the aftertreatment surface would be different. Figure 4 shows an aftertreatment inlet heat transfer model taking an inlet with insulation as an example. Heat transfer phenomena are considered in the figure including convection, conduction and radiation. According to heat energy balance, the aftertreatment skin temperature could be introduced as formula (9).

$$T_s = f(a,\ h,\ T,\ \gamma,\ T_{cat},\ a_{cat}) \tag{9}$$

where $T_s$ represents aftertreatment skin temperature; *a*, *h* and *T* stand for exhaust gas mass internal energy, enthalpy and temperature; $\gamma$ is heat transfer coefficient between the contents of different phases; $T_{cat}$ is catalyst temperature; $a_{cat}$ is heat of catalyst reaction.

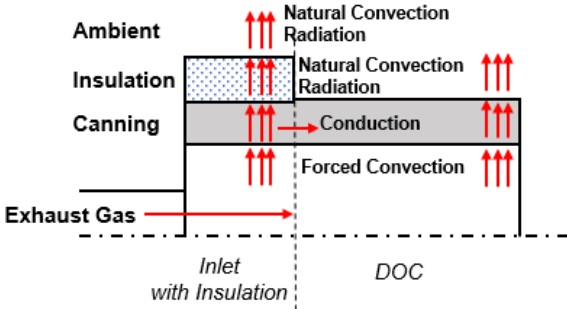

**Figure 4.** Aftertreatment inlet heat transfer model scheme.

The aftertreatment skin temperature at different points would be investigated corresponding to several kinds of experimental conditions that are carried out in this experimental part.

2.2.2. Overall Results under Different Test Conditions

Figure 5 shows a scatter diagram for all measured maximum temperatures on each test thermal mapping point under all experimental conditions with three ambient temperature environments using the two aftertreatment inlet/outlet insulation designs. From the diagram it can be seen that

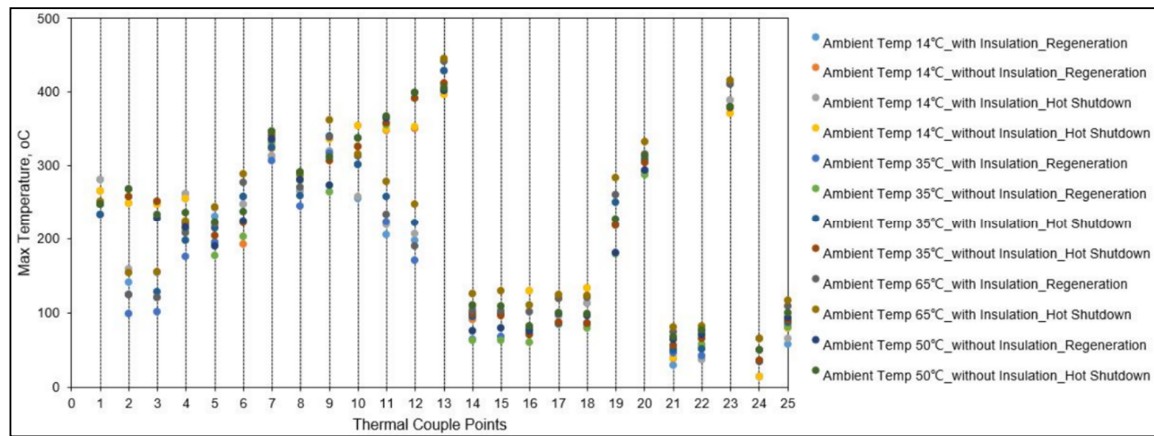

**Figure 5.** All measured temperatures scatter diagram.

1) The higher maximum temperatures concentrate on points 12, 13 and 23, which are higher than 400 °C. Point 23 is the aftertreatment outlet tube internal gas temperature. The exhaust gas comes from the engine turbo charger and flows through the catalysts for several reactions which target control strategies to meet emission requirements, which results in the exhaust gas being in a high state whichever aftertreatment inlet/outlet strategy is used. Point 12 represents the maximum skin temperature of the outlet module. From the temperature shown on point 12 under different experimental conditions, the maximum skin temperature of the outlet module is quite different depending on the experimental conditions and which is estimated according to the outlet insulation design. At point 13 it can be observed that the max skin temperature of the aftertreatment outlet tube is around 400 °C, considering different experimental conditions, which validates that the inlet/outlet insulation design has little impact on the maximum temperature of the aftertreatment outlet tube if the tube design remains the same.

2) Points 2, 3, 11 and 12 show the maximum skin temperature is distributed as a large range under different experimental conditions. These four points stand for the temperatures on the surfaces of the aftertreatment inlet and outlet modules, which means that it can be estimated that the aftertreatment inlet/outlet insulation design affects their skin temperature and needs further analysis under the experimental conditions.

3) The lower maximum temperature concentrates on points 21, 22 and 25, which all represent the surface temperature of the neighboring parts of the aftertreatment. It can be imaged that there is heat loss from the surface of the aftertreatment to its neighboring parts on the vehicle and that this results in a lower surface temperature on these parts than on the aftertreatment.

4) Points 14, 15, 16, 17 and 18 show that the maximum temperature for the DPF related skin temperature is similar for each point under different experimental conditions. This is mainly due to the fact that the DPF design is the same.

Figure 6 shows the max temperature at each thermal couple point under regeneration, and Figure 7 shows the max temperature at each thermal couple point under hot shut down. Figure 6 shows that

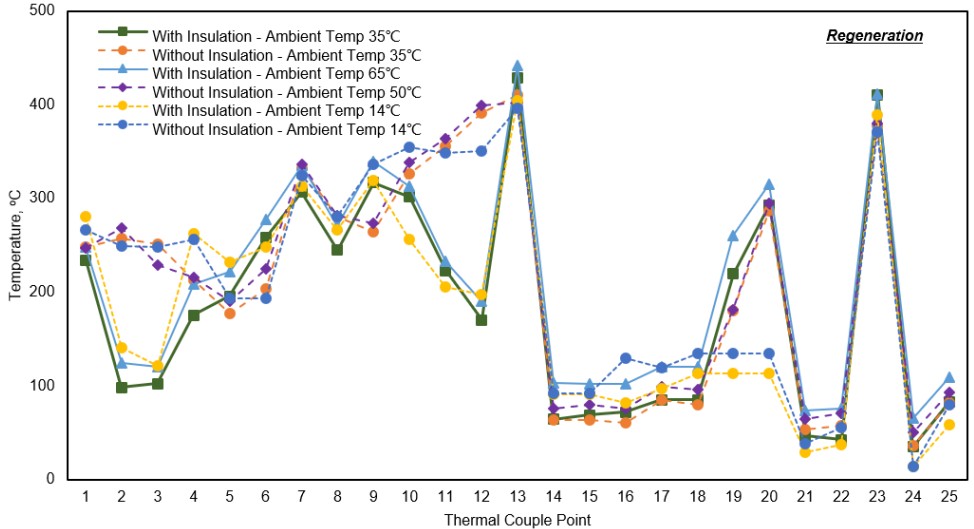

**Figure 6.** Max temperature at each thermal couple point—regeneration.

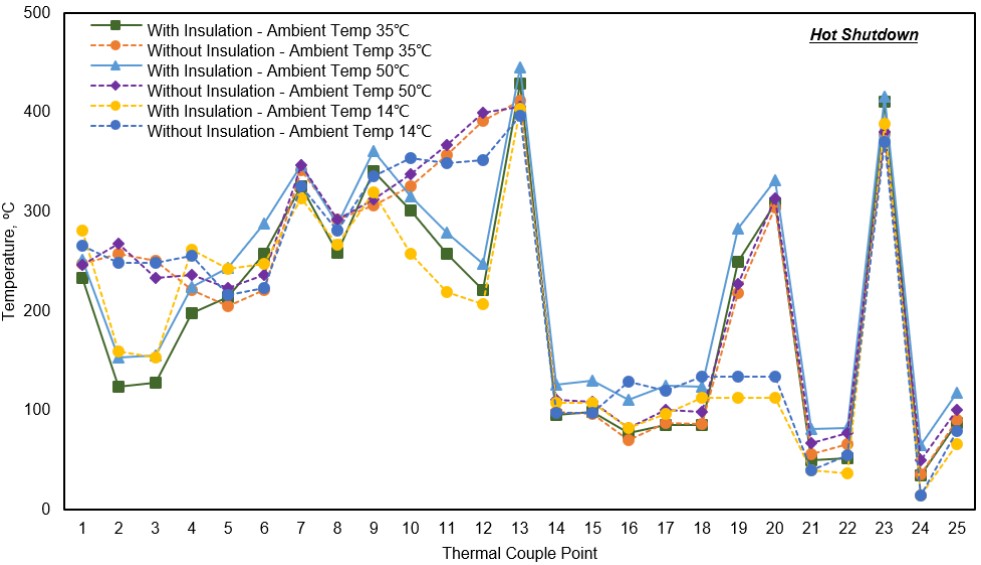

**Figure 7.** Max temperature at each thermal couple point—hot shutdown.

1) When DPF is in active generation, the internal gas temperature will be higher than 500 °C. With the design of the DPF packaging being consistent, the DPF related maximum skin temperature at points 14, 15, 16, 17 and 18 is similar at each single point under several experimental conditions. Point 17 and point 18 are the maximum skin temperatures of differential pressure sensor tubes with the highest measured data as 124 °C, which also provides guidance when choosing material for the design of differential pressure sensor tubes.

2) From the overall trend shown in this figure, it can be said that all the maximum skin temperatures rise with higher ambient temperature.

3) Per point 4, 6 and 10, it can be observed that under the same ambient temperature, the maximum skin temperature of the DOC module is similar when the aftertreatment inlet/outlet has or does not have insulation, as is the DPF module maximum skin temperature and the SCR maximum module skin temperature. In the emission experiment described above, the gas temperature at DOC_In, DPF_In, DPF_Out and SCR_Out is very similar when considering the aftertreatment inlet/outlet with or without insulation. According to the heat radiation model, the maximum skin temperatures of the catalyst modules are similar for the aftertreatment inlet/outlet with insulation and without insulation, as the packaging of the catalyst modules packaging maintains a consistent design.

4) Under the same experimental condition, point 2 and point 3 show the inlet related skin temperatures are higher for the aftertreatment inlet/outlet without insulation than for that with insulation. Point 11 and point 12 show the inlet related skin temperatures are higher for the aftertreatment inlet/outlet without insulation than for that with insulation. The inlet and outlet skin temperatures are significantly impacted by the insulation design. With insulation, the highest inlet skin temperature (point 3) is about 150 °C, and the highest outlet skin temperature (point 11) is 280 °C, while without insulation, the highest inlet skin temperature (point 3) is about 250 °C, and the highest outlet skin temperature (point 11) is 370 °C.

Figure 7 considers all the maximum skin temperatures measured under hot shut down condition, and the tested data shows the same observations in Figure 6.

### 2.2.3. Inlet/Outlet Skin Temperature Test Results and Analysis

To further investigate aftertreatment inlet and outlet skin temperature under different experimental conditions, Figure 8 shows the maximum skin temperature at aftertreatment inlet and outlet taking an ambient temperature of 35 °C as an example. Point 2 and point 3 represent aftertreatment inlet related maximum skin temperature, while point 11 and point 12 represent aftertreatment outlet related maximum skin temperature.

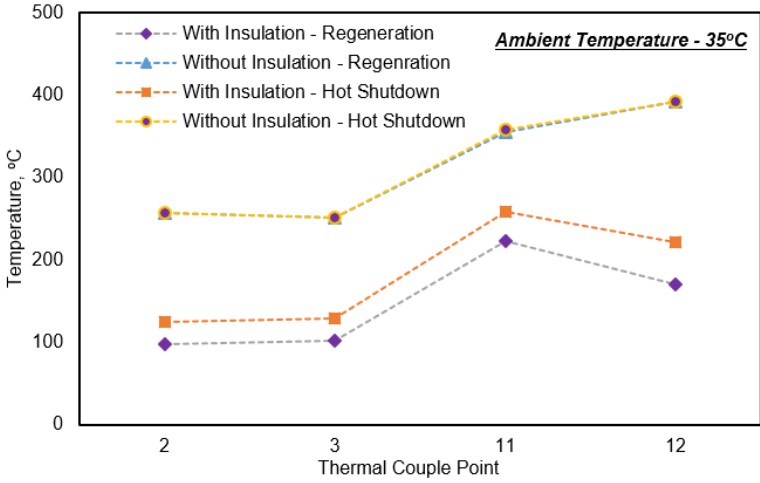

**Figure 8.** Max temperature at aftertreatment inlet and outlet surface.

1) The maximum skin temperatures of point 2 and point 3 are close to each other under the same experimental conditions with the same aftertreatment configuration, and the maximum skin temperatures of the inlet module are lower when the aftertreatment has insulation than those when the aftertreatment does not have insulation. The temperatures are higher under hot shut down condition than under regeneration condition. With insulation, the highest inlet skin temperature (point 3) is about 150 °C, while without insulation, the highest inlet skin temperature (point 3) is about 250 °C.

2) The maximum skin temperatures of point 11 and point 12 are lower for aftertreatment with insulation than those for aftertreatment without insulation. The temperatures are similar under hot shut down condition and under regeneration condition. With insulation, the highest outlet skin temperature (point 11) is about 280 °C, while without insulation, the highest inlet skin temperature (point 11) is about 370 °C.

3) The maximum skin temperature of the aftertreatment for the outlet is higher than that for the inlet for a specific aftertreatment configuration under the same experimental condition.

Generally, this could make the inlet and outlet surface maximum temperatures lower with insulation on the surface. When considering the real application on a vehicle, the aftertreatment radiated heat impact on the neighboring parts in the vehicle needs to be understood. The inlet and outlet surface temperature radiated heat impact on the neighboring parts in the vehicle is further investigated and analyzed for the real vehicle experimental cases with the two aftertreatment inlet/outlet insulation strategies. This considers the distance between the aftertreatment and its neighboring parts in the vehicle to carry out the work for the test results analysis, as well as the theoretic model explanation.

### 2.2.4. Radiated Impact on Neighboring Parts in the Vehicle

The parts in the vehicle which are close to the aftertreatment may be impacted by aftertreatment skin heat. In this nonroad vehicle arrangement, there are three parts that are around the aftertreatment—the vehicle hood near the aftertreatment inlet end plate, which is at a distance of 62 mm with point 21 to record the maximum surface temperature of the inner vehicle hood, the other side of the vehicle hood near the aftertreatment outlet end plate, which is at a distance of 62 mm with point 22 to record the maximum surface temperature of the inner vehicle hood, and a metal plate, with point 25 to record the temperature and for which the distance between point 5 and point 25 is 95 mm. For the aftertreatment inlet/outlet without insulation, the inlet/outlet end plate maximum skin temperatures are higher than those for the aftertreatment inlet/outlet with insulation. Therefore, this focuses on the radiated heat impact on the vehicle when the aftertreatment inlet/outlet does not have insulation.

Figure 9 shows the maximum surface temperature for the neighboring parts in the vehicle under regeneration condition without inlet/outlet insulation. Taking a single measured point into consideration, the temperature rises with a higher ambient temperature. When the ambient temperature is 50 °C, point 21 and point 22 measured temperatures are both lower than 90 °C, which are far from the vehicle hood temperature limitation requirements. This shows that the current distances between the aftertreatment and its neighboring parts in the vehicle are acceptable, even with the aftertreatment without insulation.

Figure 10 shows the temperature change from the aftertreatment surface to its neighboring parts in the vehicle.

1) Figure 10a shows the maximum skin temperature drops a lot from the aftertreatment inlet end plate to the vehicle hood due to the existing distance between them. Figure 10c shows the maximum skin temperature drops a lot from the aftertreatment outlet end plate to the vehicle hood due to the existing distance between them.

2) In Figure 10a, the temperature at point 2 varies under different experimental conditions. It is found that the temperature at point 21 is very similar in each condition and in a very small temperature range. This shows that the aftertreatment inlet/outlet insulation design matters little for the radiated impact on the vehicle hood with a distance of 62 mm. Even when the point 2 temperature is high at 260 °C in the experimental condition under hot shut down without inlet/outlet insulation for the

aftertreatment, the temperature at point 21 drops to 60 °C with the heat loss. The same observations can be seen in Figure 10c for the vehicle hood on the other side.

3) Figure 10a,c show the test results under a certain ambient temperature. Figure 10b shows that the heat radiated trend is the same for a specific aftertreatment configuration. Under a certain aftertreatment configuration, when the ambient temperature is higher, the temperature at point 21 is a little higher. Figure 10d shows similar analysis results. The ambient temperature plays a key role in affecting the radiated impact on the vehicle with the distance.

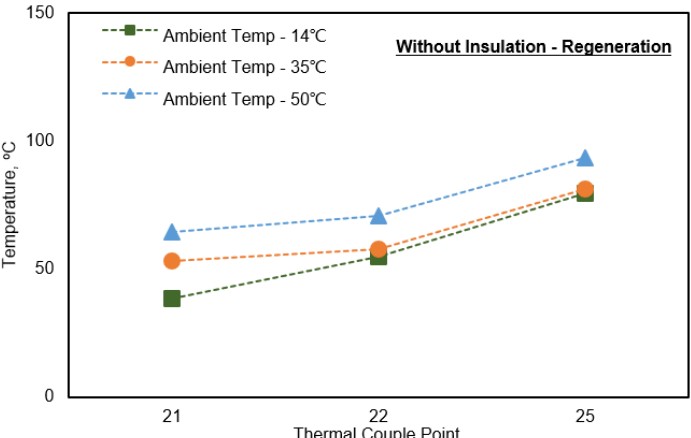

**Figure 9.** Max temperature at aftertreatment neighboring parts on vehicle.

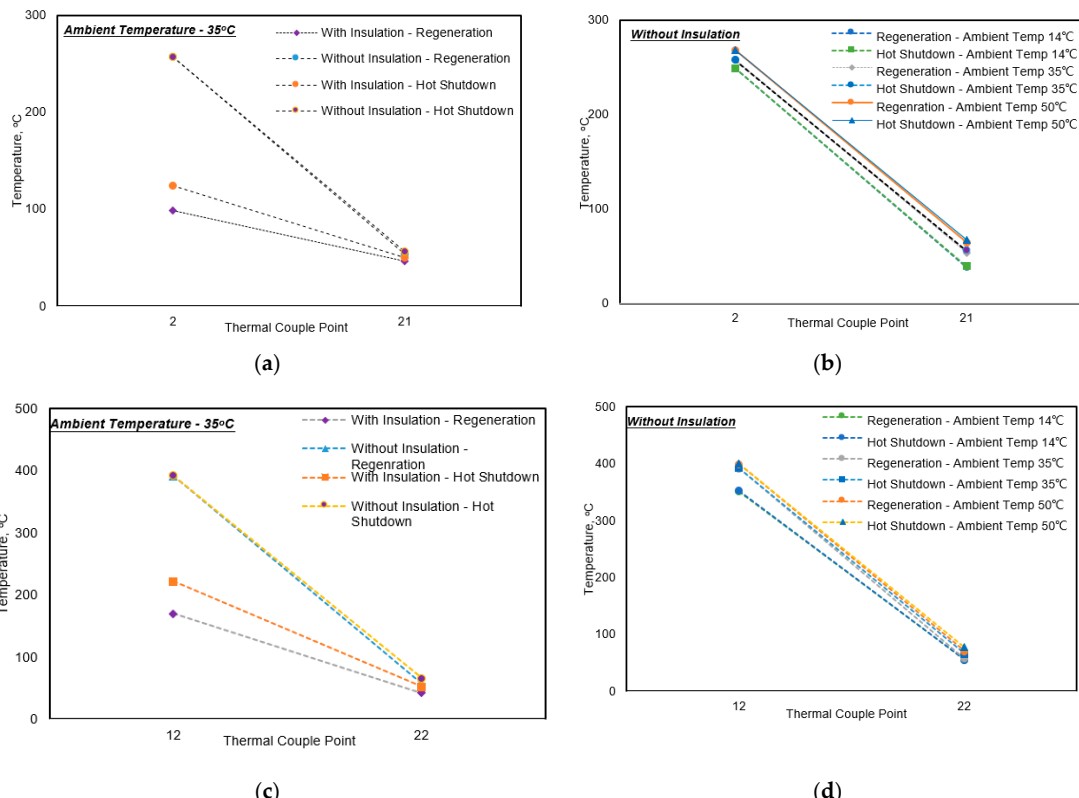

**Figure 10.** Radiated Impact from Aftertreatment on Neighboring Parts on Vehicle. (**a**) Radiated Impact from Aftertreatment Inlet to Vehicle Hood—Ambient Temperature (35 °C), (**b**) Radiated Impact from Aftertreatment Inlet to Vehicle Hood—Without Insulation, (**c**) Radiated Impact from Aftertreatment outlet to Vehicle Hood—Ambient Temperature (35 °C), (**d**) Radiated Impact from Aftertreatment outlet to Vehicle Hood—Without Insulation.

4) Whatever the aftertreatment, whether with insulation or without insulation, the aftertreatment skin temperature is high. A caution alert is needed on the aftertreatment surface so that attention is paid to safety during a service. When installing other parts around the aftertreatment system the radiated heat needs to be considered.

The radiated impact results and analysis show that in this kind of nonroad vehicle arrangement, both aftertreatment inlet/outlet with and without insulation could meet the installation requirements and their radiated heat loss makes the maximum surface temperature of the neighboring parts of the aftertreatment drop from a high aftertreatment skin temperature to meet the temperature requirements with the arranged distance. Ambient temperature also plays an important role in affecting the skin temperature.

## 3. Methodology and Experimental

### 3.1. Emission Experimental

Figure 11 shows the test set-up in this experiment. The exhaust flow enters into the aftertreatment inlet tube from the turbocharger of a 6-cylinder non-EGR(Exhaust Gas Recirculation) middle range diesel engine, and then moves across the catalysts (DOC, DPF, SCR) to the outlet tube. In this aftertreatment system, the catalysts are aged according to the catalyst coating content and the nonroad duty cycle requirements, which is 50 h under 650 °C, using the water warm method in the catalyst aging oven for all catalysts. In-cylinder post injection is used for DPF regeneration, and a doser is installed on the mixer to inject ammonia into the aftertreatment to meet a specific NOx conversion efficiency. The connection pipe between the turbocharger and the aftertreatment inlet is supposed to be 1.5 m as per nonroad application conditions. In this experimental study, the engine configuration and calibration control strategy are the same for aftertreatment with insulation for the inlet/outlet and aftertreatment without insulation for the inlet/outlet.

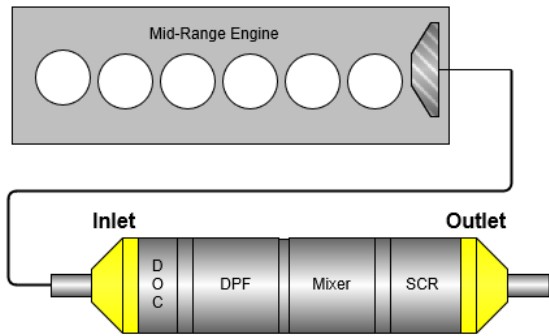

**Figure 11.** Test set-up for emission test.

### 3.2. Aftertreatment Skin Temperature and Radiated Impact Experiment

#### 3.2.1. Test Set-up

Figure 12 shows the experimental system layout of a vehicle with an aftertreatment system arranged with 21 thermal couples distributed at different points on the surface. There are three parts in the vehicle which are near to the aftertreatment surface, and three thermal couples (points 21, 22 and 25) are installed on the surface of these three parts. Point 21 is on the inner surface of the vehicle side hood at a distance of 62 mm from the aftertreatment inlet end plate, while point 22 is on the other side of the inner surface of the vehicle side hood at a distance of 62 mm from the aftertreatment outlet end plate. Point 25 is on a metal plant surface and is 95 mm from the aftertreatment surface. The ambient temperature is audited by the thermal couple on point 24. In total, 25 thermal couples are monitored for maximum temperature to analyze the aftertreatment skin temperature and its radiated heat impact on neighboring parts in the vehicle. This vehicle is powered by a 6-cylinder

non-EGR middle-range diesel engine, with the aftertreatment installed inside the vehicle hood using an additional gas tailpipe connected to the aftertreatment outlet tube. The exhaust gas which has flowed through the aftertreatment goes into the air via this tailpipe. The aftertreatment system is fastened on a mounting bracket. A detailed description of the thermal couple points is shown in Table 1.

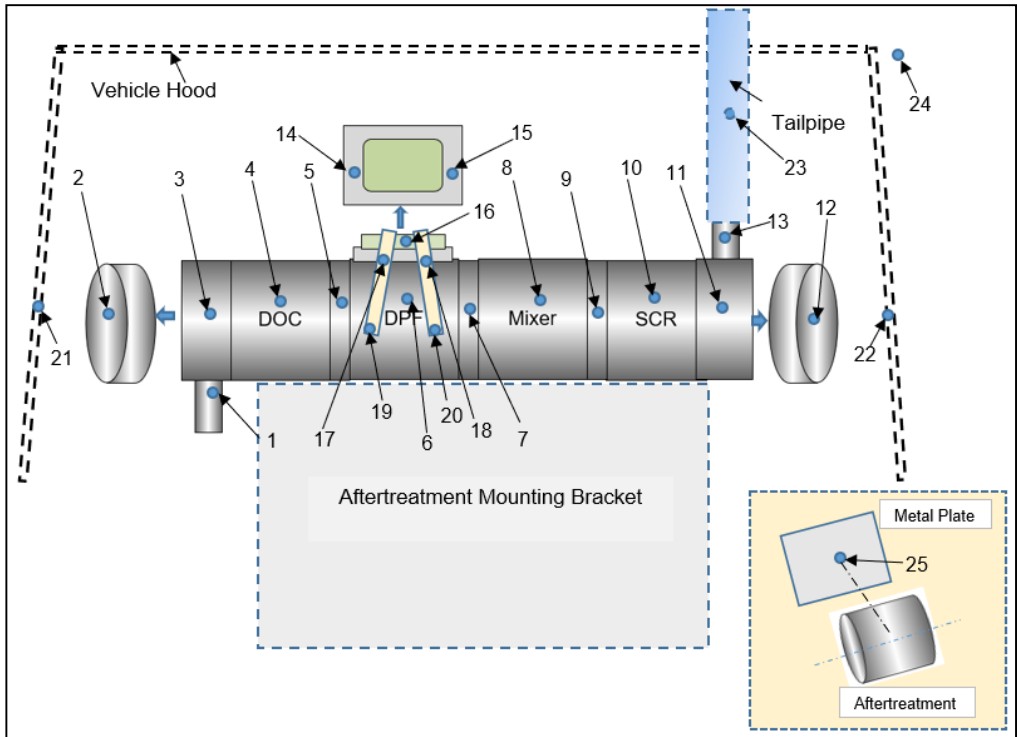

**Figure 12.** Experimental layout on vehicle.

**Table 1.** Thermal couple description.

| Thermal Couple Point | Description | Thermal Couple Point | Description |
| --- | --- | --- | --- |
| 1 | Inlet Tube Skin Temperature | 14 | Sensor Table Skin Temperature1 |
| 2 | Inlet_End Plate Skin Temperature | 15 | Sensor Table Skin Temperature2 |
| 3 | Inlet Skin Temperature | 16 | Pressure Sensor Body Skin Temperature |
| 4 | DOC Skin Temperature | 17 | Pressure Sensor Inlet Tube Skin Temperature |
| 5 | DPF_In Clamp Temperature | 18 | Pressure Sensor Outlet Tube Skin Temperature |
| 6 | DPF Skin Temperature | 19 | Pressure Sensor Tube MountingTemperature1 |
| 7 | DPF_Out Clamp Temperature | 20 | Pressure Sensor Tube MountingTemperature2 |
| 8 | Mixer Skin Temperature | 21 | Vehicle Hood Internal Skin Temperature1 |
| 9 | SCR_In Clamp Temperature | 22 | Vehicle Hood Internal Skin Temperature2 |
| 10 | SCR Skin Temperature | 23 | Outlet Internal Gas Temperature |
| 11 | Outlet Skin Temperature | 24 | Ambient Temperature |
| 12 | Outlet_End Plate Skin Temperature | 25 | Metal Plate Surface Temperature |
| 13 | Outlet Tube Skin Temperature | - | - |

### 3.2.2. Experimental Conditions Design

Data collection equipment is applied to connect with the 25 thermal couples to read all the tested temperature data under different conditions. The maximum aftertreatment skin temperature and surface temperature of the aftertreatment neighboring parts were investigated considering DPF regeneration condition and the following hot shut down condition using the two aftertreatment

inlet/outlet insulation strategies: (1) with insulation (which means the aftertreatment inlet has insulation, while the aftertreatment outlet has insulation too); (2) without insulation (which means the aftertreatment inlet does not have insulation, while the aftertreatment outlet does not have insulation either). Three ambient temperature conditions are considered in the experimental tests, 14 °C, 35 °C and 50 °C, which could be regarded as different real application environment temperatures. Figure 13 explains the design of the experimental conditions. The average ambient temperature, 50 °C, is controlled in a heat room, while 14 °C is the average ambient temperature in real natural spring and 35 °C is the average ambient temperature in real natural hot summer. The heat room temperature controlling is sensitive and achieves 65 °C when using the aftertreatment with insulation strategy.

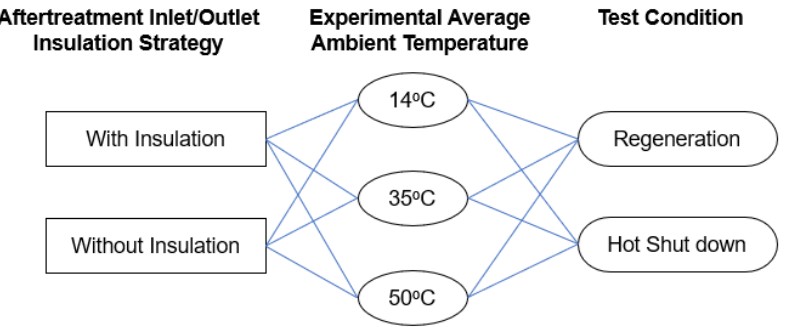

**Figure 13.** Experimental conditions design.

## 4. Conclusions

This paper presents a comprehensive experimental research work on aftertreatment skin temperature and the radiated impact on its neighboring parts in a nonroad vehicle powered by a middle range diesel engine under aftertreatment inlet/outlet with insulation and without insulation with multiple experimental conditions, as well aiming to validate the emission results with these two different aftertreatment configurations. Specially, the findings include:

1) The aftertreatment inlet/outlet with insulation and without insulation using a DOC + DPF + SCR scheme could both meet China's StageIV emission regulations and the whole vehicle arrangement.

2) The connection pipe is generally short between the aftertreatment and the engine turbo charger on this nonroad application vehicle, which results in the exhaust gas temperature of the internal aftertreatment at each point being similar for the aftertreatment inlet/outlet with insulation compared to the aftertreatment inlet/outlet without insulation. The aftertreatment skin temperature differences under these two configurations occur on the inlet module and outlet module, and the skin temperatures of the other aftertreatment modules are little impacted. These experimental results also validate the radiation model.

3) All aftertreatment skin temperatures are measured with different experimental conditions. In the future, if considering integrating other parts like sensors on the surface of the aftertreatment, configuration with insulation is recommended. As per the experimental results, the maximum inlet skin temperature can lower nearly 50% with insulation and the maximum outlet temperature could lower about 28% compared to the configuration without inlet/outlet insulation. If taking cost into consideration, the configuration without insulation is suggested. This research also introduces alternative solutions for different concerns for real applications.

4) The radiated impact on vehicle parts by the aftertreatment is similar with the two aftertreatment configurations and could meet the temperature requirements for vehicle parts.

As the radiation model is validated by the experimental results, it provides an effective reference for future applications. In the future, CFD simulation could be established using this model to monitor the temperatures with vehicle arrangement and duty cycle data before real experiments to optimize the aftertreatment packaging design accordingly.

**Author Contributions:** Formal analysis, L.X.; Investigation, L.X.; Project administration, F.Q.; Supervision, G.J.; Validation, L.X.; Writing—Original draft, L.X.; Writing—Review & editing, L.X. All authors have read and agreed to the published version of the manuscript.

**Funding:** This research was funded by Natural Science Foundation of Hubei Province (No. 2019CFB694) and Hubei Provincial Department of Education Science and Technology Research (No. B2018003).

**Conflicts of Interest:** The authors declare no conflicts of interest.

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
