# Peer review of "Research on Aftertreatment Inlet_Outlet Insulation for A Nonroad Middle Range Diesel Engine"

_catalysts, doi:10.3390/catal10040454_

Round 1

Reviewer 1 Report

·        The paper is very interesting for application reasons.

·        There is a lack of an accurate description of the research procedure and the list of research apartments with its accuracy in the work.

·        It would be advisable to show the research stand as it looked in reality.

·        Were the catalytic converter tests carried out using a real vehicle at different ambient temperatures?

·        No critical reference was made to the results obtained. What about measurement errors and uncertainties results? The authors focused on describing the advantages of their work. What are the limitations and disadvantages, each technical solution has advantages but also disadvantages or limitations?

·        In line 97 is: Exhaust gas temperature is the very important parameter when evaluating the whole aftertreatment system performance. Very important and true statement. What was the exhaust gas temperature in the examined case?

·        The description shows that the subjects were not on the road. The research mainly concerns the effect of temperature on the efficiency of the catalytic converter. The question is, has the impact of flowing air around the reactor on its efficiency been considered?

·        The data contained in Figs 9 and 10 are difficult to compare, in my opinion there is too much data. Maybe one should compare one case from Fig. 9 and the same case from Fig. 10 and compare them?

Reviewer 2 Report

The article is interestingly presented. A useful and practical study has been carried out. However, I would like to make a few shortcomings that would further improve the article:

1. Incomplete literature review. More research should be reviewed. Poor list of literature sources. Only three sources of literature are newer than five years, and all others are older. 

2. Formulas 1 and 2 are so simple and understandable to everyone that they should not be shown in research.

3. In Figure 8, the authors chose a very complex presentation of research results. It is difficult to understand their essence.

4. A very modest part of the causal analysis of the results of experimental research. It should be extended to explain the reasons and not just to establish the facts. The obtained research results should be compared with the results of other researchers.

5. The curves in Figures 11 and 13 merge with each other. They are difficult to distinguish from each other.

Reviewer 3 Report

I think that the manuscript could be published in the present form

Author Response

Appreciate your valuable time on the review. Thanks for your comments.